# Anchor Generation Optimization and Region of Interest Assignment for Vehicle Detection

**DOI:** 10.3390/s19051089

**Published:** 2019-03-03

**Authors:** Ye Wang, Zhenyi Liu, Weiwen Deng

**Affiliations:** 1State Key Laboratory of Automotive Simulation and Control, Jilin University, Changchun 130025, China; wangye13@mails.jlu.edu.cn (Y.W.); zhenyi16@mails.jlu.edu.cn (Z.L.); 2Beijing Advanced Innovation Center for Big Data and Brain Computing, Beihang University, Beijing 100191, China

**Keywords:** vehicle detection, anchor generation optimization, receptive field matching, ROI assignment

## Abstract

Region proposal network (RPN) based object detection, such as Faster Regions with CNN (Faster R-CNN), has gained considerable attention due to its high accuracy and fast speed. However, it has room for improvements when used in special application situations, such as the on-board vehicle detection. Original RPN locates multiscale anchors uniformly on each pixel of the last feature map and classifies whether an anchor is part of the foreground or background with one pixel in the last feature map. The receptive field of each pixel in the last feature map is fixed in the original faster R-CNN and does not coincide with the anchor size. Hence, only a certain part can be seen for large vehicles and too much useless information is contained in the feature for small vehicles. This reduces detection accuracy. Furthermore, the perspective projection results in the vehicle bounding box size becoming related to the bounding box position, thereby reducing the effectiveness and accuracy of the uniform anchor generation method. This reduces both detection accuracy and computing speed. After the region proposal stage, many regions of interest (ROI) are generated. The ROI pooling layer projects an ROI to the last feature map and forms a new feature map with a fixed size for final classification and box regression. The number of feature map pixels in the projected region can also influence the detection performance but this is not accurately controlled in former works. In this paper, the original faster R-CNN is optimized, especially for the on-board vehicle detection. This paper tries to solve these above-mentioned problems. The proposed method is tested on the KITTI dataset and the result shows a significant improvement without too many tricky parameter adjustments and training skills. The proposed method can also be used on other objects with obvious foreshortening effects, such as on-board pedestrian detection. The basic idea of the proposed method does not rely on concrete implementation and thus, most deep learning based object detectors with multiscale feature maps can be optimized with it.

## 1. Introduction

Vision-based advanced driver assistance system (V-ADAS) has drawn great attention from both researchers and manufacturers in recent years due to the advantages (such as affordability, large information capacity and environmentally friendly) of its camera compared with other sensors. As an important traffic participant, it is crucial for V-ADAS to detect vehicles. Early vehicle detection methods generated candidate bounding boxes roughly through knowledge-based information, such as shadows [1,2], symmetry [3,4] and vertical/horizontal edges [5,6]. After this, they classified these candidate bounding boxes through predefined feature extractors, such as Harr, Histogram of Oriented Gradient (HOG) and Gabor combined with classifiers, such as Support Vector Machine (SVM) and AdaBoost [7,8]. However, due to the complexity and variety of the scenario, these vehicle detectors can hardly obtain satisfactory performance. In recent years, due to the large number of annotated image datasets, such as Pascal [9], KITTI [10] and Cityscapes [11], and the progress of the GPU computing speed, data-driven convolutional neural networks (CNN) gained considerable attention due to its strong feature representation ability. Deep learning APIs, such as Caffe [12], TensorFlow [13] and PyTorch [14], have made the implementation of CNN easier. CNN was first used in the field of image classification and was found to have worked surprisingly well. The potential of CNN was also examined in another important computer vision research field of object detection. The earliest representative trial was Regions with CNN (R-CNN) [15]. After this, the second generation and third generation trials were faster R-CNN [16] and Faster R-CNN [17]. Each generation improved both computing speed and detection accuracy compared with the former generation.

Faster R-CNN has two stages, which are namely the region proposal stage and faster R-CNN stage. In the region proposal stage, multiscale anchors are generated uniformly at each pixel of the feature map. These anchors are used to compute the bounding box regression value and class label. The original version of Faster R-CNN generates anchors with three scales (128 × 128, 256 × 256, 512 × 512) and three aspect ratios (1:1, 1:2 and 2:1), which creates a total of nine anchors at each pixel of the feature map. The shapes of the anchors are changing but the original faster R-CNN feature extractors, such as ZF and VGG16, have only one fixed receptive field size in the last feature map. As shown in Figure 1, for a large anchor, RPN needs to make predictions using clues from a part of the anchor. In contrast, for a small anchor, useful clues may be drowned in a large number of unnecessary clues. The mismatch of the anchor sizes and the receptive field size reduces detection accuracy. The receptive field of VGG16 is relatively large for KITTI dataset, which explains why VGG16-based faster R-CNN is not good at detecting small objects.

Another problem when employing the original faster R-CNN for vehicle detection is that the uniform anchor locating method is appropriate for general-purpose object detection but is not efficient for on-board vehicle detection. To illustrate this, the KITTI dataset was employed as an example. An investigation of the relationship between the vertical central point position and the height of the 2D bounding boxes was conducted. The result shows that in the KITTI dataset, most vehicles appear in the bottom half part of images, which means that most of the anchors located at the top half part are invalid. Furthermore, due to perspective projection, the bounding box height is highly related to the vertical bounding box position. Large ground truth bounding boxes are always located at the bottom of the image and small ground truth bounding boxes are always located at the middle. A uniform anchor distribution is not efficient and accurate because large anchors located at the middle of the image and small anchors located at the bottom are all invalid.

These above-mentioned problems not only decreased detection accuracy but also increased computing time.

In order to solve these problems, the receptive fields with different scales and different aspect ratios are first generated. After this, the anchor shapes are the same as the receptive field shapes so the RPN only needs to process the appropriate number of information for classification and regression. Finally, the distribution of anchors is optimized to cover the ground truth bounding boxes more accurately without too many invalid anchors and reduce unnecessary convolutional computing.

After RPN generates many candidate ROIs, the faster R-CNN module conducts the final classification and regression. A ROI pooling layer projects an ROI to a feature map and finally forms a new feature map with a fixed size. When the multiscale feature maps are employed, there is the consideration of which feature map an ROI should be assigned to. An unreasonable ROI assignment may lead to a performance drop. For example, if a small ROI is assigned to a feature map with a large stride, the projected region in the feature map may contain too few feature pixels. After the pooling operation, it can hardly obtain enough features for final detection. This paper tries to assign ROIs more accurately to appropriate feature maps so it can obtain enough features for final detection.

The contributions of this work are as follows:(1)Different receptive fields with multishapes are generated. The anchor shapes and receptive field shapes are matched to ensure that the network could obtain the appropriate number of features.(2)The distribution of the anchor is optimized with a quantized method to reduce the number of invalid anchors. This is helpful for both accuracy and speed.(3)ROIs with different sizes are assigned to different feature maps with a new rule to ensure that the output of feature maps from the ROI pooling layer contains just the appropriate number of features for final classification and regression.

This paper is organized as follows. Previous related works will be presented in Section 2. After this, the main approach of this paper will be illustrated in Section 3. The implementation details and the results are discussed in Section 4. Finally, the conclusions and future works are given.

## 2. Related Works

In this section, the related works are reviewed. At first, the previous CNN based object detection methods are discussed in order to show the basic ideas and problems of them. After this, since the proposed method is built upon the multiscale methods, these methods also need to be reviewed. Finally, the previous rough anchor generation and ROI assignment methods are also discussed although they are not so deeply investigated in previous works.

### 2.1. CNN Based Object Detection Method

All R-CNN series object detectors have two stages, which are namely the region proposal stage and classification stage. In R-CNN, a selective research method was employed to generate proposal bounding boxes before a CNN was used as a classifier. Without any feature sharing, this results in a long computing time. This problem was partially solved in faster R-CNN. Inspired by the SPP-NET [18], faster R-CNN utilized an ROI pooling layer to project the proposal bounding boxes into the feature map of the whole image and finally obtained a fixed-size feature map. Features were shared among different proposal bounding boxes. faster R-CNN improved both computing speed and accuracy. Faster R-CNN tried to utilize the CNN-based method instead of the selective research method to generate the proposal bounding boxes. Features were shared between the region proposal stage and classification stage. The computing speed and accuracy were further improved. R-FCN [19] generated a position sensitive score map before a position sensitive ROI pooling was conducted and finally realized a bounding box classification. This method solved the conflict between shift-invariance for classification and shift-variance for detection. Some other detectors only have one stage, such as YOLO [20] and SSD [21]. Without RPN, the computing speed of YOLO and SSD is faster than Faster R-CNN. Apart for SSD, all original implementations of these detectors do not contain a multiscale property. They can hardly handle the situation when the size of object changes in a large range. SSD utilized the feature maps in different layers to obtain a multiscale property but the bottom layer feature map did not contain enough high-level semantic feature for object detection. In order to solve these problems, feature maps should have not only a multiscale property, but also high-level semantic information.

### 2.2. MultiScale Method

Among earlier researches, the SIFT [22] features were widely used for feature point matching. It preprocesses the images to different scales to form an image pyramid. Inspired by this, some CNN-based object detectors also performed preprocessing on the source image to obtain an image pyramid in the training or testing stage to obtain the scale-invariance ability [23,24]. These types of methods have obvious limitations because the computing time increases along with the number of scales. Some methods, such as SSD and MS-CNN [25], can conduct predictions in different layers with different receptive field sizes so the image pyramid was no longer needed. However, the bottom layer features with small receptive fields always did not have enough high-level semantic features. This makes it not suitable for small object detection. Some backbone networks, such as Inception [26] and DenseNet [27], concatenated features with different receptive fields to form a new feature map and let the upper layers decide which channel they use during the inferring procedure. Reference [28] utilized the DenseNet as the backbone. Its framework looks like the SSD but it performed better because of the special multiscale feature in DenseNet. Some methods, such as FPN [29] and RON [30], fused low-level and high-level features to make feature maps that not only have a multiscale ability but also contain strong semantic features. The mask R-CNN [31] and RetinaNet [32] used the FPN-based method to generate multiscale features. All these above-mentioned methods generate square receptive fields, which are appropriate for general-purpose object detection. However, in the case of vehicle detection, due to the special shape of vehicle, receptive fields with different aspect ratios will be better.

### 2.3. Anchor Generation Method and ROI Assignment

The original faster R-CNN generated nine anchors with different scales and aspect ratios uniformly at each pixel of the last feature map. It can be seen as a method of exhaustion to ensure that every object can be covered. When using FPN together with Faster R-CNN, anchors were generated at different scale feature maps. The number of anchors increased due to the large size of the feature map, which leads to a slow computing speed. In the SSD object detector, the situation is similar. This uniform anchor generation method was designed for general-purpose objection. When using this in the on-board vehicle detection, many anchors are invalid. Furthermore, anchor shapes did not coincide with receptive field shapes, which may lead to low detection accuracy. To date, according to our knowledge, no study has focused on optimizing anchor generation for the on-board applications.

After the region proposal stage, ROI pooling layer projects each ROI to a feature map and forms a fixed-size feature map for final classification and regression. When multiscale feature maps are induced, there is the consideration of which feature map an ROI should be assigned to. The Original FPN based faster R-CNN assigns ROI only according to its size, without taking the feature stride into consideration. The situation is similar in MS-CNN and many other object detectors with multiscale feature maps. This ROI assigning method is not accurate enough. The number of features in the extracted feature map may be not suitable for final prediction. According to our knowledge, no study has focused on accurate ROI assignment.

## 3. Approach

In this section, the proposed method is introduced. At first, in order to deal with vehicles of different sizes, multishape receptive fields are generated with a network specially designed for vehicle detection. After this, the anchor generation method is optimized in a quantified way to match the receptive field shapes and reduce the number of invalid anchors. Finally, an accurate ROI assigning method is proposed in order to ensure that the extracted feature maps after ROI pooling could contain the appropriate number of features.

### 3.1. MultiShape Receptive Field Generation

The proposed method is built upon Faster R-CNN, which is a famous general-purpose object detector. There are three subnetworks in Faster R-CNN, which are namely the feature extractor, RPN header network and faster R-CNN header network. At first, the feature extractor takes a single image as the input and outputs for the feature maps. After this, RPN header network takes the feature maps as the input and outputs for the candidate ROIs. Finally, faster R-CNN header network takes the candidate ROIs and the feature maps as inputs and outputs for the final detection results. This is shown in Figure 2.

The RPN in the original faster R-CNN assumes that there is a class of anchors with different shapes in only one feature map with one fixed receptive field shape (171 and 228 pixels for ZF and VGG). It can hardly handle the situation when the anchor sizes changes within a large range. To solve this problem, FPN is employed to generate multiscale feature maps with both the multiscale property and high-level semantic features. The FPN feature extractor is shown in Figure 2a. In FPN, a set of layers with the same size feature maps is called a stage. The last layers of these stages are used to generate the feature pyramid. The VGG16 network is used as a backbone feature extractor and has five stages. The last four stages are employed to generate the feature pyramid. The first stage is not employed because its receptive field is too small and the size is too large, which means too many anchors will be located in this feature map. We denote the last layers of these stages as C2, C3, C4, C5 and their corresponding pyramidal features as P2, P3, P4, P5. The ith feature map Pi is a sum of the up-sampled feature map of Pi+1 which has higher level semantic information and convolutional feature map Ci, which has a smaller receptive field size. FPN outputs feature maps with proportional sizes and different receptive fields. The top-down pathway and lateral connections creates the feature maps with both different receptive fields and high-level semantic features. The computing speed of FPN is also satisfactory because the added up-sampling, 1 × 1 convoluting and element-wise adding do not involve too many computations. Furthermore, both low-level features and high-level features are utilized efficiently. The main procedure is similar when using other backbone networks, such as RESNET [33], and is expected to obtain better performance.

The combination of the VGG16 backbone network, FPN and Faster RCNN can be seen as a baseline. They are not the key point of the proposed method. All detectors with multiscale feature maps could employ this method. VGG16 can be changed to RESNET, FPN can be changed to MS-CNN and even Faster R-CNN can be changed to SSD. The basic idea in the following paragraphs does not change. This paper just focuses on the performance improvement compared with the baseline detector.

The receptive fields are square in the original FPN-based faster R-CNN because all shapes of convolutional kernels in FPN feature extractor are square. However, the shapes of most vehicle bounding boxes are rectangular and rectangular receptive fields are more useful for vehicle detection. In order to obtain rectangular receptive fields, rectangular convolutional kernels are needed. However, it is not reasonable to implement rectangular convolutional kernels in the FPN feature extractor because they are not appropriate for the second stage faster R-CNN prediction. As shown in Figure 2b, instead of implementing them in the FPN feature extractor, rectangular receptive fields are induced by using three convolutional kernels with different shapes at the end of the RPN header network. In consideration of the vehicle shape and pose, most of the bounding boxes do not have large aspect ratios so the convolutional kernel sizes are set as 1 × 1, 1 × 7 and 1 × 13. In this implementation, there are four feature maps with different scales generated by FPN and three different aspect ratios generated by the three branches in RPN header network. Thus, twelve different receptive field shapes can be obtained. Their shapes are shown in Table 1. As an example, we give an intuitive explanation of the receptive field shape on P2. As shown in Figure 3, in our implementation, the receptive field size R2 of feature map P2 is equal to 18 and the feature stride FS2 is equal to 2. The receptive field sizes of final three different branches are equal to 18 × 18, 18 × 30 and 18 × 42. In original FPN based faster R-CNN, there is only one receptive field size in P2, which is equal to 18 × 18.

The implementation of this idea in other baseline detectors is easy. For example, MS-CNN and SSD utilize feature maps in different layers to induce the multiscale property, which corresponds to FPN in the proposed method. It is intuitive to apply this method to them, which will only require a change in the final layers with three different branches that have different kernel shapes.

As a summary, in this section, a network is constructed, which can generate receptive fields with various scales and various aspect ratios. Various scales of receptive fields are induced by FPN and various aspect ratios are induced by different convolutional kernel shapes in the RPN header network. The various receptive field shapes in this method are appropriate for the special application of on-board vehicle detection and allow us to achieve further optimization in next section.

### 3.2. Anchor Generation Optimization

First, in order to remove the mismatch of the receptive field and anchor, the anchor shapes are made to be the same as the receptive field shape as shown in Figure 3. Twelve different receptive field shapes generated in the former section correspond to twelve different anchor shapes and they are assigned to different feature maps and different branches presented in the former section. After locating the anchors at the center point of each feature map pixel, twelve anchor distributions can be obtained.

In both the original faster R-CNN and FPN-based faster R-CNN, the anchors are located uniformly at each pixel of feature maps. After this, the class labels and location regression labels are assigned to all anchors. Only two types of anchors are regarded as positive samples: (1) anchors with the highest Intersection overlap Union (IoU) with a ground truth bounding box; and (2) anchors with an IoU higher than 0.7 with a ground truth bounding box. Using this method for defining positive samples, for each anchor distribution, only a certain part of them have the possibility of being sampled as a positive sample and too many anchors are invalid in vehicle detection. The situation is even worse for the FPN-based faster R-CNN because the number of anchors is too high in the feature map with a large size, such as P2. Due to the perspective projection effect, the bounding box size has an approximately linear relationship with its vertical position. This can be seen as a prior knowledge in the scenario of vehicle detection and can be utilized for anchor distribution optimization. Inspired by this, a quantified method is proposed to determine which part of the anchor distribution is valid.

At first, an approximately linear relationship between bounding box vertical position and its height is necessary. As shown in Figure 4, according to the triangle similarity, the following equation can be derived.
(1)$$\frac{\left(v\left(H_{b}\right)-\frac{H}{2}\right)\;\times \;\rho }{h-\frac{H_{v}}{2}}=\frac{f}{d}=\frac{H_{b}\;\times \;\rho }{H_{v}}$$
where h is the height of the camera from the ground; H is the height of the image; $H_{v}$ denotes the average height of the vehicles in the real world; d is the distance of the vehicle to the camera; f is the focal length; and ρ is the size of each pixel. From Equation (1), a mean relationship between the height of the bounding box $H_{b}$ and the vertical position v can be obtained
(2)$$v\left(H_{b}\right)=\frac{h-\frac{H_{v}}{2}}{H_{v}}\;\times {\;H}_{b}+\frac{H}{2}$$

Equation (2) corresponds to the central green line in Figure 5, which indicates the mean relationship of these two values.

However, the real vertical position v changes around the mean value due to many disturbances. The minimum and maximum boundaries of v need to be determined. At first, the relative pitch angle between the camera and the ground plane is taken into consideration. Given the maximum relative pitch angle α between the camera and the ground plane, two boundary equations to determine the maximum and minimum values of v can be derived:(3)$$v_{\min}\left(H_{b}\right)=\frac{h-\tan\alpha \;\times \;d-\frac{H_{v}}{2}}{H_{v}}\;\times {\;H}_{b}+\frac{H}{2}$$
(4)$$v_{\max}\left(H_{b}\right)=\frac{h+\tan\alpha \;\times \;d-\frac{H_{v}}{2}}{H_{v}}\;\times {\;H}_{b}+\frac{H}{2}$$
where $d=\;\frac{{\mathrm{fH}}_{v}}{{\rho H}_{b}}$ according to triangle similarity. After simplifying Equations (3) and (4), these relationships can be obtained:(5)$$v_{\min}\left(H_{b}\right)=\frac{h-\frac{H_{v}}{2}}{H_{v}}\;\times {\;H}_{b}-\frac{\tan\alpha \;\times \;f}{\rho }+\frac{H}{2}$$
(6)$$v_{\max}\left(H_{b}\right)=\frac{h-\frac{H_{v}}{2}}{H_{v}}\;\times {\;H}_{b}+\frac{\tan\alpha \;\times \;f}{\rho }+\frac{H}{2}$$

After this, the variation of the height of vehicle is taken into consideration. $\delta_{v}$ is the maximum variation value of the height of vehicle. The equations become:(7)$$v_{\min}\left(H_{b}\right)=\frac{h-\frac{H_{v}+\delta_{v}}{2}}{H_{v}+\delta_{v}}\;\times {\;H}_{b}-\frac{\tan\alpha \;\times \;f}{\rho }+\frac{H}{2}$$
(8)$$v_{\max}\left(H_{b}\right)=\frac{h-\frac{H_{v}-\delta_{v}}{2}}{H_{v}-\delta_{v}}\;\times {\;H}_{b}+\frac{\tan\alpha \;\times \;f}{\rho }+\frac{H}{2}$$

Equations (7) and (8) correspond to two orange lines in Figure 5.

For the ith feature map $P_{i}$, $R_{i}$ represents its receptive field height. Bounding boxes with height $H_{b}$ in the range of:(9)$$\frac{R_{i-1}+R_{i}}{2}<H_{b}<\frac{R_{i}+R_{i+1}}{2}$$ are more reasonable to be inferred with $P_{i}$. The minimum and maximum vertical positions in the source image coordinate of the responsible area for $P_{i}$ are:(10)$${\mathrm{vmin}}_{\mathrm{pi}}={\;v}_{\min}\left(\frac{R_{i-1}+R_{i}}{2}\right)$$
(11)$${\mathrm{vmax}}_{\mathrm{pi}}={\;v}_{\max}\left(\frac{R_{i}+R_{i+1}}{2}\right)$$

These two values correspond to the purple lines in Figure 5. The meaning of these two values is that most bounding boxes with the height in the range of Equation (9) are located in the range from ${\mathrm{vmin}}_{\mathrm{pi}}$ to ${\mathrm{vmax}}_{\mathrm{pi}}$ in the image coordinates and they are more reasonable to be inferred in feature map $P_{i}$ because its receptive field shape (also the anchor shape) is more similar to the bounding box shapes. In other words, it is more reasonable to use the feature map $P_{i}$ for inferring the bounding boxes located in the region from ${\mathrm{vmin}}_{\mathrm{pi}}$ to ${\mathrm{vmax}}_{\mathrm{pi}}$ in the image coordinate so only a part of $P_{i}$ that corresponds to this region in the image coordinate is useful. Furthermore, the anchors are located only in this useful part. After this, they are mapped from the source image coordinate to the feature map in order to finally determine which part of $P_{i}$ is meaningful for object detection.
(12)$${\mathrm{vfmin}}_{\mathrm{pi}}=\frac{{\mathrm{vmin}}_{\mathrm{pi}}}{{\mathrm{FS}}_{\mathrm{pi}}}$$
(13)$${\mathrm{vfmax}}_{\mathrm{pi}}=\frac{{\mathrm{vmax}}_{\mathrm{pi}}}{{\mathrm{FS}}_{\mathrm{pi}}}$$
where ${\mathrm{FS}}_{\mathrm{pi}}$ is the feature stride of $P_{i}$ and only the region from ${\mathrm{vfmin}}_{\mathrm{pi}}$ to ${\mathrm{vfmax}}_{\mathrm{pi}}$ is employed for final classification.

The aforementioned parameter values are shown in Table 2 and the results of the valid area of each feature map in the source image coordinate are shown in Figure 6.

With this method, only a certain part of each feature map is employed to perform classification and regression. The number of convolutional computation is reduced and the number of invalid anchor is also reduced.

However, the basic idea of anchor distribution optimization can be easily transplanted to other baseline detectors. For two stage detectors, such as MS-CNN, nothing changed because it also locates anchors in the different feature maps in a similar way to faster R-CNN. For one stage object detector, such as SSD, the concept of “default bounding box” is very similar to the concept of “anchor” in Faster RCNN and they are also located in different feature maps. We need to just match different receptive field shapes to different anchor shapes or different default bounding box shapes before removing invalid anchors or invalid default bounding boxes using the same principle.

As a summary, the purpose of this section is to determine the anchor shape and optimize anchor distribution. In order to remove the mismatch, the anchor shapes are made to be the same as the receptive field shapes. After this, the perspective effect is utilized as prior knowledge to determine which part of each anchor distribution is valid. The shape matching is helpful to improve the detection accuracy and the distribution optimization is helpful to improve computing speed.

### 3.3. ROI Assignment

The former section is aimed at optimizing RPN. As shown in Figure 2c, after the region proposal stage, the ROIs generated from RPN are fed into the faster R-CNN header network together with the feature maps for final prediction. The ROI Pooling layer of the R-CNN header network projects an ROI to the corresponding region in the feature map according to the stride of the feature map, which then divides this region into a fixed number of sections (7 × 7 in this implementation). After this, the max-pooling operation is employed on every section, which finally forms a fixed size feature map. Finally, the feature map will go through some fully connected layers for the final prediction. As mentioned above, four feature maps with different feature strides are generated with FPN. There is the consideration of which feature map an ROI should be assigned to. If a small ROI is assigned to a feature map with a large stride, the size of the corresponding region in the feature map may be too small. When divided into many sections, the feature size in each section may be less than one pixel and the information in it is not enough for final classification and regression. Otherwise, if a large ROI is assigned to a feature map with a small stride, too many pixels are contained in one section and it is bad for the max-pooling computing speed. These situations are shown in Figure 7. In this section, a new method is proposed to determine which feature map an ROI should be assigned to.

Given the width $W_{r}$ and height $H_{r}$ of an ROI, its equivalent size is computed as follows:(14)$$S_{e}=\;\sqrt{W_{r}\;\times {\;H}_{r}}$$

After this, compute the expected equivalent stride:(15)$$\mathrm{ES}=\;\frac{S_{e}}{2.0\;\times {\;S}_{\mathrm{ROIPooling}}}$$
where $S_{\mathrm{ROIPooling}}$ is the size of the output feature map after ROI pooling. In this implementation, $S_{\mathrm{ROIPooling}}$ was equal to 7. The expected equivalent stride ES can be regarded as a appropriate feature stride of this ROI for pooling.

ROIs are assigned according to ES and the feature stride of each feature map. An ROI should be assigned to a feature map with a feature stride that is closest to its ES. For the ith feature map Pi, ROI with ES in this range should be assigned to it
(16)$$\frac{{\mathrm{FS}}_{\mathrm{pi}-1}+{\mathrm{FS}}_{\mathrm{pi}}}{2}<\mathrm{ES}<\frac{{\mathrm{FS}}_{\mathrm{pi}}+{\mathrm{FS}}_{\mathrm{pi}+1}}{2}$$
where ${\mathrm{FS}}_{\mathrm{pi}}$ is the feature stride of Pi.

With this strategy, large ROIs are assigned to feature maps with large strides and small ROIs are assigned to feature maps with small strides. This method could match the ES of ROI and the ${\mathrm{FS}}_{\mathrm{pi}}$ of feature map more accurately, which ensures that the feature maps after ROI pooling layer contain an appropriate number of information for final classification and regression. This would enhance the accuracy of detection.

## 4. Experiment and Discussion

In this section, the validity of the proposed method is demonstrated. At first, the implementation details are presented. After this, a comprehensive experiment setup is employed to show the effectiveness of every method presented in this paper and make sure that the comparison is on the same baseline. The experiment results are also discussed in detail.

### 4.1. Implementation Details

The proposed method is implemented with PyTorch 0.4, an open source deep learning framework developed by Facebook AI Research and accelerated with CUDA 8.0 and cuDNN 5.0. The employed GPU is Titan XP.

The KITTI dataset is employed for training and testing.

The combination of VGG16 backbone network, FPN and faster R-CNN is employed as a baseline object detector. The proposed method could also be useful for other baseline detectors.

RPN is trained with two stages. In the first stage, the entire RPN is trained and in the second stage, only RPN header network is finetuned. Each stage runs 20 epochs. One image is feed to the network in each step. The definitions of the foreground and background anchor are same as that in the original faster R-CNN. A total of 256 samples are chosen randomly for each image. The learning momentum is 0.9 and the learning rate is 0.001.

When training the faster R-CNN, the feature extractor in RPN is employed and only the faster R-CNN header network is fine-tuned. The faster R-CNN training has only one stage. This stage runs 20 epochs. Two images are fed to the network at each step. The ROIs with IoUs of more than 0.5 are defined as positive samples. The ROIs with IoUs less than 0.1 are defined as negative samples. A total of 64 samples are chosen randomly for each image (128 for the two images). The learning momentum is 0.9 and the learning rate is 0.001.

In order to allow for fairer comparisons, all these hyper-parameter settings and training pipeline are the same for all experiments.

### 4.2. Experiment Setup and Result Discussion

At first, the improvement of RPN with the proposed method needs to be shown. Three experiments with different region proposal methods are compared. The first experiment uses VGG16 without FPN as the feature extractor of RPN. Only the last feature map is used to locate anchors and generate region proposals, which is similar to the original version of faster R-CNN. The second experiment uses VGG16 with FPN as feature extractor, with four feature maps employed to locate the anchors and generate region proposals. This method is similar to the original FPN. The third experiment is the proposed method, which creates multishape receptive fields, anchor-receptive field shape matching and anchor distribution optimization.

In fact, the computing speed of some postprocessing methods in the faster R-CNN is highly related to the number of anchors as a smaller anchor number usually leads to faster computing speed. The first experiment has the smallest anchor numbers. Anchors are located only in the last feature map with a small feature size. Due to its simple network framework and small anchor number, it saves more time from both network running and postprocessing and has the fastest computing speed. The second experiment generates anchors uniformly on every feature map. The anchor number is large because of the presence of large feature maps, such as P2. The network is also more complicated than the first experiment and the computing speed is low. The proposed method utilizes only a part of each feature map to locate anchors and perform inference. Compared with the second experiment, the anchor number is reduced and both the running time of the network and postprocessing time are lower.

The average recall (AR) proposed in [34] is used to evaluate the accuracy of RPN. For the first experiment, only one receptive field size is not adequate for the multiscale vehicle detection, especially for small vehicles. This experiment obtained the lowest AR. When combined with FPN in the second experiment, better results are obtained. Multiscale feature maps play an important role in vehicle detection. Improvement can be observed especially in small vehicles. When optimized with the proposed method in the third experiment, the AR is higher than the second experiment despite the reduced anchor number. This result is due to many reasons. The addition of rectangle convolution kernels induces rectangle receptive fields, which is suitable for vehicle detection. The consistency of the anchor shape and receptive field shape also makes the classification and regression more accurate. The results of these experiments are shown in Table 3.

After this, the effectiveness of the proposed ROI assigning method in the faster R-CNN needs to be proved. Average precision (AP) is employed to evaluate the performance. There are also three experiments. The first experiment has no special ROI assignment and all the ROIs are assigned to P5 feature map. The second experiment uses the ROI assigning method in the original paper of FPN. The third experiment uses the proposed method. In order to exclude the influence of the different RPNs, the same RPN is used in these three experiments. For one image, these experiments generate same region proposals so the ROI assigning method can be evaluated more fairly. The first experiment achieves the lowest AP. This is mainly because the stride of P5 is too big, especially for small ROIs, and ROI pooling layer cannot generate sufficient information for final classification and regression. In the second experiment, the situation gets better. The ROI assignment in the original FPN paper can also assign large ROIs to feature maps with a large stride and small ROIs to feature maps with a small stride in a similar way to the proposed method. However, it is not accurate enough. In the third experiment, the proposed method gets the highest AP. With this method, each feature map after ROI pooling contains just appropriate number of features. The results of these experiments are shown in Table 4.

Finally, the performance of the whole method needs to be evaluated. There are also three experiments. The first experiment is the original faster R-CNN without any additional improvement. The second experiment uses the FPN as the feature extractor. The region proposal method and ROI assigning method are the same as that in the original FPN paper without rectangle receptive field and anchor generation optimization. The third experiment is the proposed method. The proposed method obtains the highest AP. Further, compared with the original FPN-based faster R-CNN, the proposed method gains an improvement in computing speed. The results of these experiments are shown in Table 5. The precision–recall curves of the three experiments are shown in Figure 8.

Some example detection results are shown in Figure 9.

## 5. Conclusions and Future Work

In this paper, we present a method to optimize the faster R-CNN, especially for on-board vehicle detection. Methods, including multishape receptive field generation, anchor generation optimization and ROI assignment, are employed to improve the performance. Multishape receptive fields allow the detector to detect vehicles with different sizes and shapes. Anchor generation optimization eliminates the mismatch between the anchor shape and the receptive field shape, which reduces the number of invalid anchors. ROI assignment makes the number of features after ROI pooling more suitable for final prediction. The proposed method is evaluated on the KITTI dataset and shows a significant improvement in both computing speed and detection accuracy. Future works will focus on other vehicular computing vision applications, such as vehicle 3D bounding box detection and state estimation.

## Figures and Tables

**Figure 1 sensors-19-01089-f001:**
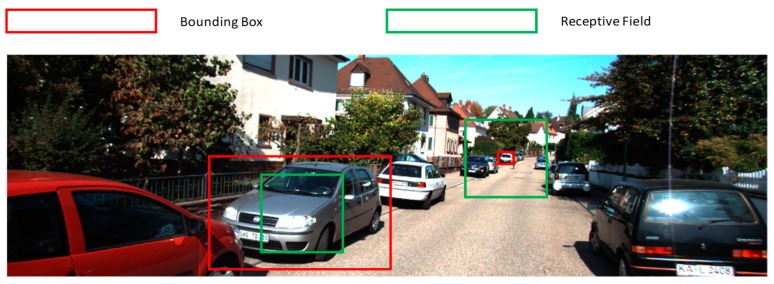
Mismatch of the receptive field and the vehicle bounding box.

**Figure 2 sensors-19-01089-f002:**
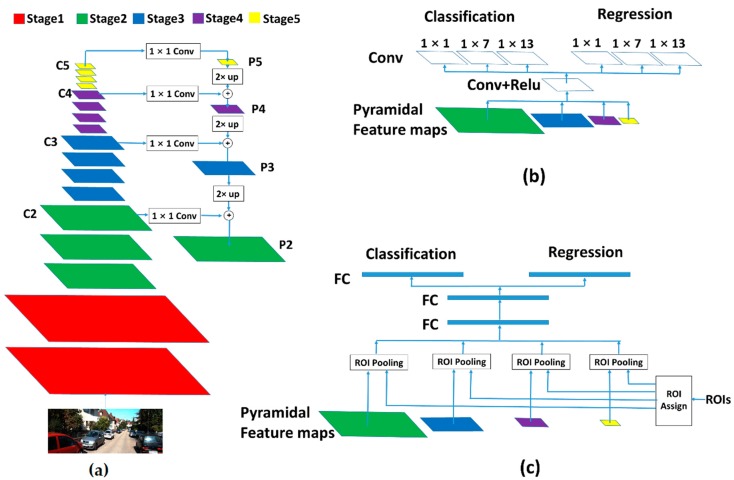
Three subnetworks in the proposed method. The feature extractor based on FPN is shown in (**a**) and multiscale feature maps can be generated with it. The header network of RPN is shown in (**b**), with rectangular convolutional kernels on the top of it. Combined with FPN feature extractor, multishape receptive fields can be generated. The header network of faster R-CNN is shown in (**c**).

**Figure 3 sensors-19-01089-f003:**
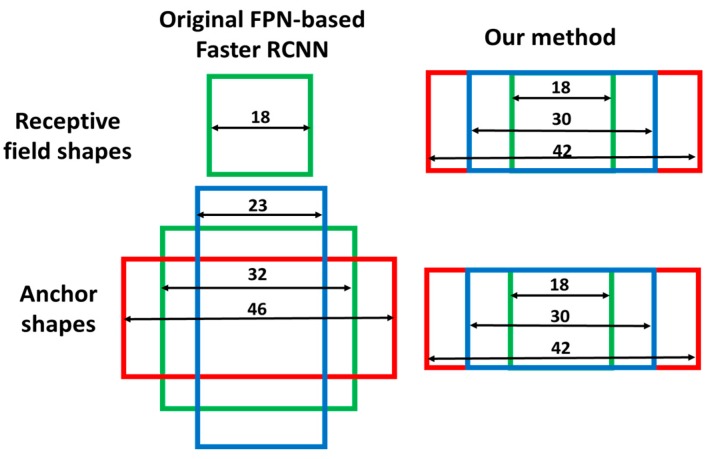
Different receptive field shapes and anchor shapes in P2.

**Figure 4 sensors-19-01089-f004:**
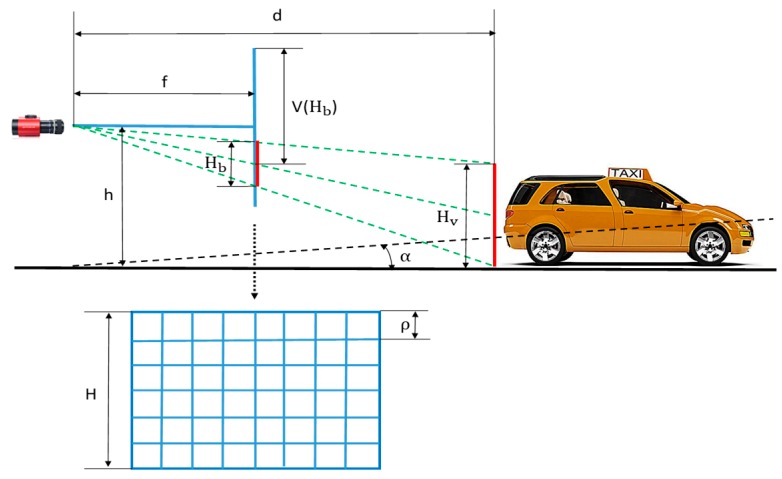
Illustration of the values in equations.

**Figure 5 sensors-19-01089-f005:**
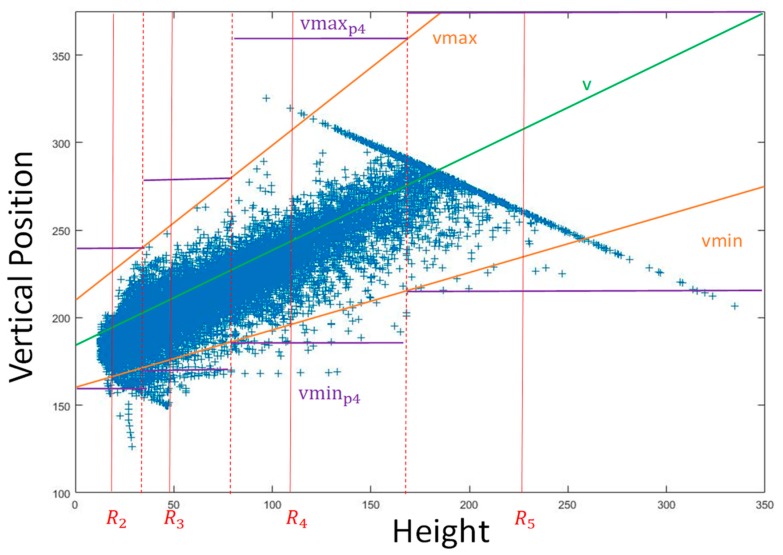
Our method to determine which part of the image a feature map should be involved in the region proposal stage.

**Figure 6 sensors-19-01089-f006:**
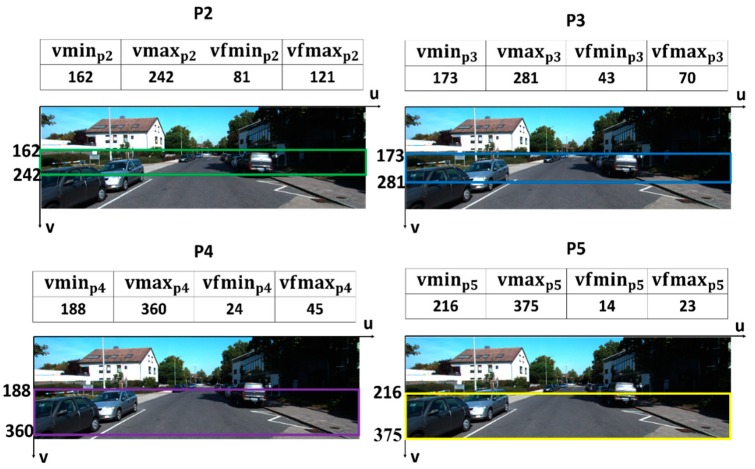
Computing results and valid region of each feature map.

**Figure 7 sensors-19-01089-f007:**
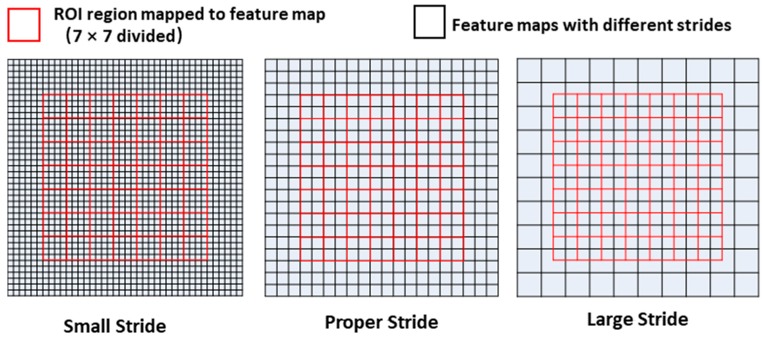
Appropriate feature stride for ROI pooling.

**Figure 8 sensors-19-01089-f008:**
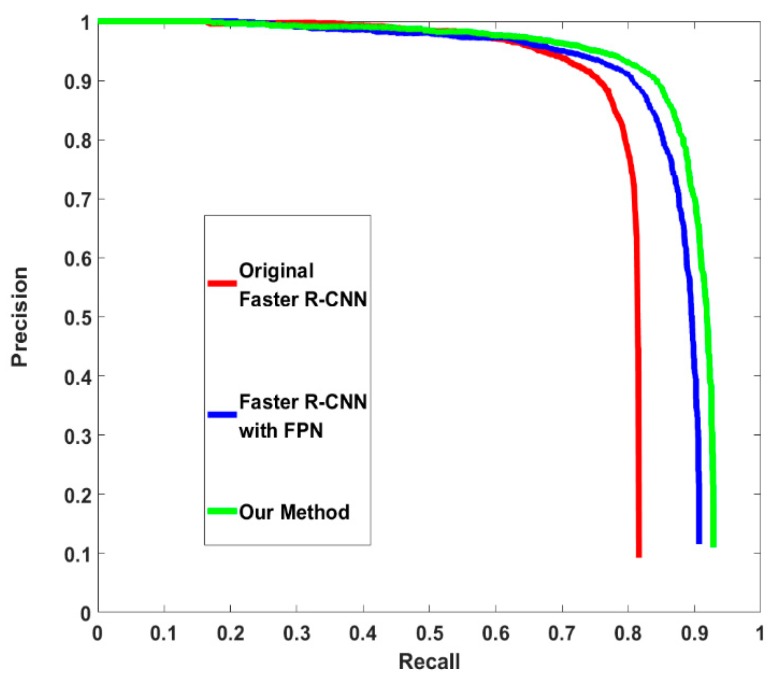
Precision–recall curves of different detectors.

**Figure 9 sensors-19-01089-f009:**
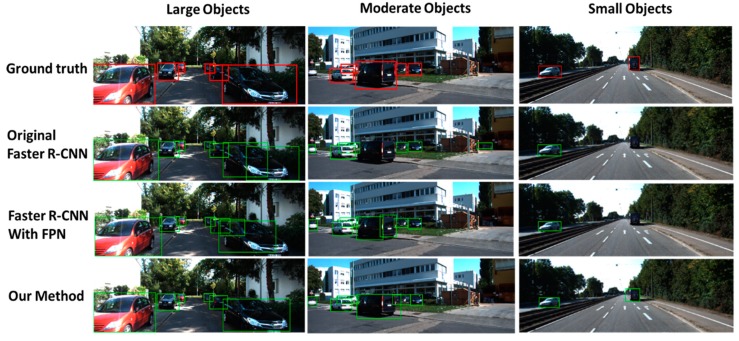
Some examples of the detection results of different vehicle detectors on Kitti.

**Table 1 sensors-19-01089-t001:** Twelve receptive fields with different shapes. The anchor shape is the same with corresponding receptive field. R and FS are the receptive field size and stride of each feature map.

	P2 Feature Map	P3 Feature Map	P4 Feature Map	P5 Feature Map
1 × 1 kernel	R2 × R2 = 18 × 18	R3 × R3 = 48 × 48	R4 × R4 = 108 × 108	R5 × R5 = 228 × 228
1 × 7 kernel	R2 × (R2 + (7 − 1) × FS2) = 18 × 30	R3 × (R3 + (7 − 1) × FS3) = 48 × 72	R4 × (R4 + (7 − 1) × FS4) = 108 × 156	R5 × (R5 + (7 − 1) × FS5) = 228 × 324
1 × 13 kernel	R2 × (R2 + (13 − 1) × FS2) = 18 × 42	R3 × (R3 + (13 − 1) × FS3) = 48 × 96	R4 × (R4 + (13 − 1) × FS4) = 108 × 204	R5 × (R5 + (13 − 1) × FS5) = 228 × 420

**Table 2 sensors-19-01089-t002:** Parameter values in this section.

H.	h	f/ρ	H_v_	δ_v_	α
375	1.65 m	721.54	1.6 m	0.4 m	2°

**Table 3 sensors-19-01089-t003:** Performance of RPN in different experiments.

Experiments	Anchor Number	Computing Time	AR	AR_S	AR_M	AR_L
1 (Original RPN)	15 K	0.011 s	0.288	0.213	0.296	0.330
2 (RPN with FPN)	463 K	0.023 s	0.352	0.348	0.356	0.344
3 (Proposed method)	142 K	0.016 s	**0.370**	**0.362**	**0.371**	**0.373**

**Table 4 sensors-19-01089-t004:** Performance of ROI assignments in different experiments.

Experiments	AP	AP_S	AP_M	AP_L
1 (Without ROI assignment)	0.762	0.403	0.793	0.834
2 (Assignment in Original FPN)	0.849	0.868	0.833	0.856
3 (Proposed method)	**0.857**	**0.877**	**0.849**	**0.858**

**Table 5 sensors-19-01089-t005:** Performance of final detection results in different experiments.

Experiments	Computing Time	AP	AP_S	AP_M	AP_L
1 (Original Faster R-CNN)	0.037 s	0.785	0.632	0.790	0.865
2 (Faster R-CNN with FPN)	0.07 s	0.849	0.868	0.833	0.856
3 (Proposed method)	0.055 s	**0.872**	**0.888**	**0.872**	**0.866**

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
