# Peer review of "Anchor Generation Optimization and Region of Interest Assignment for Vehicle Detection"

_sensors, 2019, doi:10.3390/s19051089_

Round 1
Reviewer 1 Report
The article is quite interesting and seems to present better results (generally) than other methodologies analyzed using the same dataset. In general, it can be considered for publication in the journal, subject to correction of the following issues:
Minor issues:
Line 105: Please, describe the structure of section 2 (and please revise the font of section 2 title).
Line 105: Section 2 seems too weak, it could be greatly improved.
Line 145: Sections 3, 4 and 5 should be unified in a single section called "Proposal" or something similar.
Sections 3, 4 and 5: Some aspects of the proposal need to be better explained, such as the functioning of Figure 2.
Line 299: Please, introduce section 6 before the first subsection.
Line 376: The section should describe the conclusion, not a summary of the previously presented content.
Line 387: Any proposal to face this new line?
There are a lot of format issues, such as:
Title: The use of abbreviations (ROI) should be avoided in the title.
Lines 5-7: Please, unify the affiliation of the authors.
Line 10: Please, define R-CNN.
Line 21: Please, define ROI.
Line 40: Please, unify the format in references (space missing before [4])
Line 41: Please, include blank space before references ("edges[5],[6]"). Revise and unify the format of all the references included in the article.
Line 42: Please, define HOG and SVM.
...
This part must be greatly improved before being published.
Author Response
1. Line 105: Please, describe the structure of section 2 (and please revise the font of section 2 title).
Thanks for your suggestions. A paragraph is added in section 2 in describe the structure of it, and the font is also revised.
2. Line 105: Section 2 seems too weak, it could be greatly improved.
Thanks for your suggestions. Some more contents about previous ROI assigning methods are added in this section.
3. Line 145: Sections 3, 4 and 5 should be unified in a single section called "Proposal" or something similar.
Thanks for your suggestions, but I feel a little confused about this. In fact, section 3, 4, 5 are not all about region proposal. Section 3 is about network architecture. Section 4 is about anchor generation. Section 5 is about ROI assignment. Only Section 3 and section 4 are related to region proposal. I unified these three sections in a single section called “Approach”, and a brief introduction paragraph of our method is added at beginning of this section. I think this will be better.
4.Sections 3, 4 and 5: Some aspects of the proposal need to be better explained, such as the functioning of Figure 2.
Thanks for your suggestions. Some more descriptions are added in these sections.
5.Line 299: Please, introduce section 6 before the first subsection.
Thanks for your suggestions. A introduce paragraph is added in “Experiment and Discussion” section.
6.Line 376: The section should describe the conclusion, not a summary of the previously presented content.
Thanks for your suggestions. Some contents in this section is modified.
7.Line 387: Any proposal to face this new line?
Vehicle 3D bounding box detection is much more complicated than 2D bounding box detection. We try to combine monocular depth estimation and 2D bounding box detection to achieve this goal. This work is still in progress.
There are a lot of format issues, such as:
8.Title: The use of abbreviations (ROI) should be avoided in the title.
Thanks for your suggestions. The title is changed to “Anchor generation optimization and region of interest assignment for vehicle detection”
9.Lines 5-7: Please, unify the affiliation of the authors.
Thanks for your suggestions. The affiliation of the authors is unified.
10.Line 10: Please, define R-CNN.
Thanks for your suggestions. “R-CNN” is defined in the revised manuscript.
11.Line 21: Please, define ROI.
Thanks for your suggestions. ROI is defined in the revised manuscript.
12.Line 40: Please, unify the format in references (space missing before [4])
Thanks for your suggestions. This kind of issues are all modified.
13.Line 41: Please, include blank space before references ("edges[5],[6]"). Revise and unify the format of all the references included in the article.
Thanks for your suggestions. This kind of issues are all modified.
14.Line 42: Please, define HOG and SVM.
Thanks for your suggestions. HOG and SVM are defined in the revised manuscript.
Reviewer 2 Report
The paper is well written, the proposal is sound and the results are convincing.
I have no major concerns in relation to this paper and it can be published almost as is. My very minor suggestions are:
1) The authors should proofread the paper to eliminate typos that can be found here and there along the text.
2) Please, refrain from using the first person, “we, our, us” along the text.
3) There are some constructions that are a bit weird, such as the caption of Figure 2: "Figure 2. The main framework of our method. (a) is the feature extractor based on FPN..." The sentence in the letter "a" should not start with a verb, which is also valid for the other letters.
Author Response
1) The authors should proofread the paper to eliminate typos that can be found here and there along the text.
Thanks for your suggestions. This is because English is not my native language. I feel sorry about these typos, and I try my best to eliminate them in the revised manuscript.
2) Please, refrain from using the first person, “we, our, us” along the text.
Thanks for your suggestions. This kind of issues are modified.
3) There are some constructions that are a bit weird, such as the caption of Figure 2: "Figure 2. The main framework of our method. (a) is the feature extractor based on FPN..." The sentence in the letter "a" should not start with a verb, which is also valid for the other letters.
Thanks for your suggestions. This kind of issues are modified.